# The PAGES CoralHydro2k Seawater $\delta^{18}O$ Database: A FAIR-aligned compilation of seawater $\delta^{18}O$ data to uncover 'hidden' insights from the global ocean

Alyssa R. Atwood[1], Andrea L. Moore[1], Kristine L. DeLong[2], Sylvia E. Long[1], Sara C. Sanchez[3], Jessica A. Hargreaves[4], Chandler A. Morris[5], Raquel E. Pauly[1], Émilie P. Dassié[6], Thomas Felis[4], Antje H.L. Voelker[7], Sujata A. Murty[8], Kim M. Cobb[5]

[1]Dept. of Earth, Ocean and Atmospheric Science, Florida State University, Tallahassee, FL, 32303, USA
[2]Geography and Anthropology and the Coastal Studies Institute, Baton Rouge, LA, 70803, USA
[3]Atmospheric and Oceanic Science Department, University of Colorado Boulder, Boulder, CO, 80309, USA
[4]MARUM - Center for Marine Environmental Sciences, University of Bremen, Bremen, 28359, Germany
[5]Department of Earth, Environmental, and Planetary Sciences, Brown Univ., Providence, RI, 02912, USA
[6]UMR EPOC, University of Bordeaux, Pessac, 33615, France
[7]Divisão de Geologia e Georecursos Marinhos, Instituto Português do Mar e da Atmosfera, Alges, 1495-165, Portugal
[8]Atmospheric and Environmental Sciences, University at Albany, State University of New York, Albany, NY, 12226, USA

*Correspondence to*: Alyssa R. Atwood (aatwood@fsu.edu)

**Abstract.** The stable isotope values of seawater ($\delta^{18}O$ and $\delta^{2}H$) provide valuable information on the exchange of water between the ocean, atmosphere, and cryosphere and on ocean mixing processes. As such, observational seawater $\delta^{18}O$ and $\delta^{2}H$ data place powerful constraints on hydrologic changes in the modern ocean. Seawater $\delta^{18}O$ data are also essential for calibrating paleoclimate proxies based on the $\delta^{18}O$ of marine carbonates and are an increasingly critical diagnostic tool for assessing model performance and skill in isotope-enabled global climate models. Despite their broad value, no centralized and actively-curated database for this type of data exists, even though a growing number of new seawater $\delta^{18}O$ datasets have been generated over the last decade. As such, many seawater $\delta^{18}O$ datasets remain 'hidden'. To improve the accessibility of seawater $\delta^{18}O$ data for the Earth Science research community, the Past Global Changes (PAGES) CoralHydro2k project has created a new, machine-readable, and metadata-rich database of observational seawater $\delta^{18}O$ data, paired with seawater $\delta^{2}H$ and salinity data, that is compliant with findability, accessibility, interoperability, and reusability (FAIR) standards for digital assets. The data has been collected from public databases and repositories, direct researcher data submissions, scientific papers, and student theses. In total, the PAGES CoralHydro2k Seawater $\delta^{18}O$ Database contains over 18,600 data points with extensive metadata that makes the database suitable for a myriad of research applications. For hidden data, we searched for and included all datasets within the global ocean. For public data, our data collation efforts were focused on the upper 50 m from 35°N to 35°S (to aid in CoralHydro2k's seawater $\delta^{18}O$ reconstruction studies using $\delta^{18}O$ and Sr/Ca in tropical-subtropical coral skeletons). We also provide a set of best practices to the community for reporting seawater isotope data in the future.





**Short Summary.** The stable isotopic composition of seawater is a valuable tool for studying the global water cycle in the past, present, and future. However, an active repository dedicated to archiving this type of data has been lacking, and many datasets remain hidden from public view. We have created a new database of observational seawater isotope data that is rich in metadata, publicly accessible, and machine readable to increase its availability and usability for a variety of Earth Science applications.

# 1 Introduction

## 1.1 Progress and challenges in the synthesis of seawater oxygen isotope data

The stable isotopes of water ($\delta^{18}$O and and $\delta^2$H) are powerful tracers of the global water cycle, tracking water as it continuously cycles between the ocean, atmosphere, and land. As water molecules undergo phase changes during this process, the lighter, more abundant isotope ($^{16}$O and $^1$H) is preferentially vaporized during evaporation with respect to the heavier, less abundant isotope ($^{18}$O and $^2$H), while $^{18}$O and $^2$H are preferentially condensed during precipitation (Dansgaard, 1964). This partitioning of isotopes based on mass allows the isotope values (where $\delta^{18}$O $= \left[ \frac{\frac{^{18}O}{^{16}O}sample}{\frac{^{18}O}{^{16}O}standard} - 1 \right] * 1000$) of water to be used as a tracer of the hydrologic cycle (Dansgaard, 1954; Galewsky et al., 2016; Gat, 1996). In the ocean, the isotope values of seawater ($\delta^{18}$O$_{sw}$ and $\delta^2$H$_{sw}$) can provide valuable information on an array of processes, including heat and mass exchange with the atmosphere (*via* precipitation and evaporation), large-scale ocean circulation, and freshwater input from rivers and ice sheets (Akhoudas et al., 2021; Benetti et al., 2016; Benway and Mix, 2004; Biddle et al., 2019; Craig and Gordon, 1965; Dee et al., 2023; Frew et al., 2000; Imbrie et al., 1984; Jacobs et al., 1985; Lisiecki and Raymo, 2005; Meredith et al., 1999; Strain and Tan, 1993). $\delta^{18}$O$_{sw}$ and $\delta^2$H$_{sw}$ values can also provide insight into other ocean tracers such as salinity, since they covary strongly due to the influence of evaporation and precipitation on each of these variables (Craig and Gordon, 1965; LeGrande and Schmidt, 2011). However, because key processes act differentially on salinity as compared to the stable isotope values, $\delta^{18}$O$_{sw}$ and $\delta^2$H$_{sw}$ provide an additional degree of freedom for constraining ocean mixing and the local moisture budget. Furthermore, stable isotope measurements, such as $\delta^{18}$O in marine biominerals and $\delta^2$H in lipids, can be used to trace plankton and animal movement and provide provenance for ecology, conservation, archaeology, and food forensics studies (Doubleday et al., 2022). Given these wide-ranging applications, seawater isotope data are used in a wide range of fields, including oceanography, atmospheric science, geology, marine biology, archaeology, and geography.

Seawater isotope values also create a common unit that uniquely links paleoclimate reconstructions to modern climate observations and isotope-enabled model simulations. Modern $\delta^{18}$O$_{sw}$ data are essential for the calibration of paleoclimate proxies of past ocean variability based on the $\delta^{18}$O of marine carbonates such as corals, foraminifera, mollusks, ostracods, and coralline algae. Recent paleoclimate data assimilation efforts such as the Last Millennium Reanalysis project (e.g., Tardif et al., 2019) would greatly benefit from a spatial network of $\delta^{18}$O$_{sw}$ data to improve quantification of proxy uncertainty and for



training the proxy system models that underlie those efforts. Modern $\delta^2H_{sw}$ data are used in the calibration of
paleoceanographic proxies based on the $\delta^2H$ of alkenones and other lipid biomarkers in marine sediments (e.g., Eglinton and
Eglinton, 2008). When used in tandem with $\delta^{18}O$ data (i.e., to calculate d-excess in surface ocean and overlying water vapor),
these data can be used to constrain evaporation parameters (e.g., Benetti et al., 2014). As such, observational and reconstruction
efforts based on seawater isotope values enable scientists to better understand the underlying physics that govern the water
cycle, and to extend hydroclimate records back to the preindustrial era, thus contextualizing anthropogenic climate change and
improving the skill of future climate projections.

Observational $\delta^{18}O_{sw}$ data can also be used to provide boundary conditions in climate models and to assess model performance
and skill. The increasing integration of oxygen isotopes of water in climate models – from models of intermediate complexity
to fully coupled Earth System Models (e.g., Blossey et al., 2010; Bong et al., 2024; Bony et al., 2008; Brady et al., 2019;
Cauquoin et al., 2019; Dee et al., 2015; Field et al., 2014; Fiorella et al., 2021; Kurita et al., 2011; Lee and Fung, 2008; Noone
and Simmonds, 2002; Nusbaumer et al., 2017; Risi et al., 2010, 2020, 2021; Schmidt et al., 2007; Tada et al., 2021; Wei et al.,
2018; Werner et al., 2011; Yoshimura et al., 2008) – bolsters the interpretation of modern and paleoclimate observations, while
also providing opportunities to test model performance in resolving key features of the hydrologic cycle, e.g., the representation
of moisture transport, circulation, and surface water fluxes.


Paralleling recent advances in the numerical simulation of water isotopes, new analytical capabilities have also developed in
recent years, including new *in situ* atmospheric measurement techniques and strategies (Finkenbiner et al., 2022; Gupta et al.,
2009; Henze et al., 2022), and the development of global atmospheric data products from a variety of remote sensors (e.g.,
Diekmann et al., 2021; Schneider et al., 2022; Worden et al., 2019). As a result, measurements of water isotopes have become
increasingly incorporated in coordinated observing networks and monitoring studies of precipitation and atmospheric water
vapor, including the Global Network of Isotopes in Precipitation ([www.iaea.org/services/networks/gnip](http://www.iaea.org/services/networks/gnip)) and the National
Ecological Observatory Network ([www.neonscience.org/](http://www.neonscience.org/)).

However, no such coordinated observing network for seawater $\delta^{18}O$ currently exists. Unlike meteorological observations on
land, observations of ocean hydrological properties (e.g., precipitation, evaporation, and salinity) are either limited to the past
few decades *via* satellite remote sensing and the ARGO program (Wong et al., 2020) or are confined to select coastal and
island locations that have the necessary infrastructure to support sustained *in situ* measurements of ocean surface properties.
Furthermore, these ocean observations rarely include $\delta^{18}O_{sw}$ because there is currently no cost-effective, easily deployable
instrumentation to measure seawater isotopes *in situ.* Thus seawater samples must be taken back to a laboratory for isotopic
analysis. Despite these structural challenges, a growing number of $\delta^{18}O_{sw}$ datasets have been generated in recent decades due
to the accelerated collection of $\delta^{18}O_{sw}$ samples, new instrumentation such as cavity-ring down isotope analyzers with reduced



analytical costs, and the capability to measure both $\delta^{18}O$ and $\delta^2H$ in parallel, and new sampling devices that enable long-term seawater sample collections (e.g., Jannasch et al., 2004; Khare et al., 2021).

In recognition of the broad value of $\delta^{18}O_{sw}$ data to the Earth Sciences, a major effort to gather $\delta^{18}O_{sw}$ data occurred in the 1990s (Bigg and Rohling, 2000; Schmidt, 1999; Schmidt et al., 1999) and resulted in the development of the NASA's Goddard Institute for Space Studies (GISS) Global Seawater Oxygen-18 database (https://data.giss.nasa.gov/o18data/), which contains over 26,000 global measurements of $\delta^{18}O_{sw}$ (and some $\delta^2H$ data) from the 1950s to 2000s. In 2006, that database was used to construct a global gridded dataset of $\delta^{18}O_{sw}$ and to characterize regional relationships between $\delta^{18}O_{sw}$ and salinity (LeGrande

and Schmidt, 2006) and it has subsequently been used in a broad range of studies involving $\delta^{18}O_{sw}$. However, the NASA GISS database is no longer actively updated, with the last $\delta^{18}O_{sw}$ measurement added in 2011. As a result, a growing number of new $\delta^{18}O_{sw}$ datasets published since 2011 remain without an active $\delta^{18}O_{sw}$-specific data repository in which to archive the data. Researchers have instead provided the $\delta^{18}O_{sw}$ data in the supplemental tables of journal articles, or have archived the $\delta^{18}O_{sw}$ data with other geochemical data (e.g., coral $\delta^{18}O$), in data repositories such as the National Centers for Environmental

Information (NCEI) for Paleoclimatology (https://www.ncei.noaa.gov/products/paleoclimatology) and Pangaea (https://www.pangaea.de/). Because these datasets can be difficult to find, non-machine-readable, and/or decentralized, they are not easily accessible to the wide range of research communities that would benefit from this data (see a related review by Chamberlain et al., 2021). Furthermore, many publishers and several funding agencies now require researchers to archive their data in FAIR and public repositories. For these reasons, a comprehensive database of $\delta^{18}O_{sw}$ data that is publicly available and

actively maintained is critically needed.

### 1.2 The PAGES CoralHydro2k Seawater $\delta^{18}O$ Database

Inspired by the PAGES (Past Global Changes) Hydro2k Workshop in 2016 (PAGES Hydro2k Consortium, 2017), the PAGES CoralHydro2k project was formed in 2017 to investigate the variability of hydrology and temperature in the tropical surface ocean during the past 2000 years based on the combination of coral $\delta^{18}O$, which varies with temperature and $\delta^{18}O_{sw}$, and the

strontium-to-calcium ratio (Sr/Ca) in corals, which is a temperature proxy. The CoralHydro2k project was built upon previous PAGES 2k efforts, namely Ocean2k and Iso2k (Konecky et al., 2020, 2023; Tierney et al., 2015), which compiled published coral $\delta^{18}O$ records and other data into new machine-readable databases to track temperature and hydroclimate changes over the Common Era. To aid in the calibration and interpretation of the paired coral $\delta^{18}O$ and Sr/Ca records in the database, and derive coral-based reconstructions of seawater $\delta^{18}O$, the CoralHydro2k project also started to compile $\delta^{18}O_{sw}$ data.


In recognition of the broad value of $\delta^{18}O_{sw}$ data and the growing number of $\delta^{18}O_{sw}$ datasets that have been generated during the last two decades, the CoralHydro2k Seawater $\delta^{18}O$ Database project was launched in 2020 to recover 'hidden' $\delta^{18}O_{sw}$ data that were not easily findable. During the past five years, we have integrated these records, along with any associated $\delta^2H$, salinity, and temperature data, with data from public databases and repositories to create a new, centralized, machine-readable,



and metadata-rich database that aligns with findability, accessibility, interoperability, and reusability (FAIR) standards (Wilkinson, 2016). Here we provide a detailed description of the PAGES CoralHydro2k Seawater $\delta^{18}$O Database. We highlight the opportunities and limitations of this database, and provide a set of best practices to the community for reporting this type of seawater isotope data in the future.

## 2 Methods

### 2.1 Collaborative model

CoralHydro2k and its Seawater $\delta^{18}$O Database started in 2017 as a project in Phase 3 of the PAGES 2k network, a long-running initiative to study past global changes over the last 2000 years and to compile paleoclimate data in publicly available, machine-readable databases (PAGES 2k Network Coordinators, 2017). CoralHydro2k included team members from the Phase 1 PAGES 2k Ocean2k working group (Tierney et al., 2015) and Phase 2 Iso2k working group (Konecky et al., 2020) and many new

members, particularly from the coral paleoclimate community. CoralHydro2k continues into Phase 4 of PAGES 2k, focusing on reconstructing past changes in tropical ocean temperature and hydroclimate using paired Sr/Ca and $\delta^{18}$O from coral archives over the last 2000 years (Hargreaves et al., 2020; Walter et al., 2023). Recurring calls went out within the international paleoclimate community for working group members, coral experts, and paleo data assimilation experts to join the effort with monthly teleconference meetings and one in-person meeting in 2019 (Hargreaves et al., 2020). As a result, the CoralHydro2k

database was produced, a global, actively curated compilation of coral $\delta^{18}$O and Sr/Ca proxy records of tropical ocean hydrology and temperature for the Common Era (Walter et al., 2022, 2023). A number of sub-projects were developed in conjunction with CoralHydro2k, including a project to develop new proxy system models (PSM) for coral $\delta^{18}$O. The group working on this sub-project realized that the spatial and temporal coverage of observational $\delta^{18}O_{sw}$ data were too sparse to integrate into the PSM framework and that many new $\delta^{18}O_{sw}$ datasets produced during the last few decades are not easily

findable or accessible.

Therefore, CoralHydro2k formed a new sub-project in 2020 to compile existing seawater $\delta^{18}$O data with rich metadata following FAIR standards (Atwood et al., 2024; DeLong et al., 2022). Researchers were invited to submit their data to the CoralHydro2k Seawater $\delta^{18}$O Database *via* a Qualtrics survey and accompanying YouTube video that provided instructions

on how to submit data. Additionally, the team set up a Seawater Oxygen Isotopes Community (https://www.earthchem.org/communities/seawater-oxygen-isotopes/) in the EarthChem Library (ECL), a data repository that archives, publishes, and provides access to data in the geosciences. The ECL offers a suite of services for data preservation and access, including long-term archiving and data registration with a Digital Object Identifier (DOI). Through the new Seawater Oxygen Isotopes Community, new seawater $\delta^{18}$O (and $\delta^2$H) datasets can be submitted and assigned a DOI, which

allows the datasets to be cited and tracked when used by other researchers. The CoralHydro2k members promoted this new



database at international conferences in the United States and Europe, in the PAGES newsletter (Atwood et al., 2024), and in Eos, the monthly magazine of the American Geophysical Union (AGU) (DeLong et al., 2022).

The workload for assembling the seawater data and metadata was performed by CoralHydro2k members and new members of the Seawater $\delta^{18}$O Database sub-project. The team was made of volunteer scientists from all academic levels, including undergraduate and graduate students, postdoctoral researchers, and early- to senior-level scientists from a number of international academic and research institutions. The work was completed remotely in synchronous working sessions and asynchronously across several virtual platforms (Google Suite, Slack, and Zoom). Data discovery, metadata protocols, and compilation were done collaboratively as the project progressed.

**2.2 Data aggregation and formatting**

The CoralHydro2k Seawater $\delta^{18}$O Database was designed to be as inclusive and comprehensive as possible in its record-selection criteria to support the project's goal of developing a FAIR database of global seawater $\delta^{18}$O measurements, paired with $\delta^2$H and salinity measurements, and to include as much 'hidden' data as possible. Thus, the Seawater $\delta^{18}$O Database selection criteria were less restrictive than other PAGES 2k efforts, and the database includes data from peer-reviewed 175 scientific literature, student theses and dissertations, public data repositories, and direct author submission.

In alignment with FAIR data principles, the Seawater $\delta^{18}$O Database contains extensive metadata. Where available, the seawater $\delta^{18}$O data is paired with seawater $\delta^2$H, salinity, and temperature data. Eight metadata fields are required, with an additional 44 optional metadata fields that provide supporting information on site information, sample collection and storage 180 notes, the isotope analysis method and instrumentation, and error information. Additionally, a template is provided to assist researchers with future submissions to the database and to establish a set of best practices for reporting seawater isotope data.

For hidden data, we searched for and included datasets spanning all depths and all latitudes across the global ocean. For publicly available data, we focused on including data from the upper 50 m between 35ºN to 35ºS (to aid in CoralHydro2k's 185 seawater $\delta^{18}$O reconstruction studies using $\delta^{18}$O and Sr/Ca in tropical-subtropical corals).

**2.3 Metadata description and quality control**

The metadata is described in this section and Tables 1–2. The CoralHydro2k Seawater $\delta^{18}$O Database team implemented several rounds of quality control measures for the data and metadata. Following the Iso2k database procedure (Konecky et al., 2020), each metadata field has an associated quality control certification "Level" from 1 to 6, described below and in Table 1. 190 Level 1 and Level 2 metadata fields constitute 'essential' metadata, and if a dataset lacked one of these fields, it was excluded from the database.



- *Level 1* fields are required for inclusion in the database and they contain standardized vocabularies, according to Table 2. They are recommended as primary fields for filtering and querying records in the database. They were subject to the highest Quality Control (QC) standard. Examples of Level 1 metadata are: "Collection year", "Collection month", "Latitude", "Longitude", and "Depth".

- *Level 2* metadata fields are required for inclusion, but they are not generalizable enough to use standardized vocabularies. They were subject to the highest QC standard and the metadata were obtained from the original publication or data source. An example of Level 2 metadata is "Site name or geographic area".

- *Level 3* metadata fields add important supplementary information related to the seawater $\delta^{18}O$ measurements. They contain standardized vocabularies and can be used as secondary fields for filtering and querying the database; however, they are generally not available for all records and thus not required for inclusion in the database. They were subject to the highest QC standard. Examples of Level 3 metadata are: "Collection day", "$\delta^{18}O$ error", "$\delta^{18}O$ analysis technique", "Water isotope analysis date", "$\delta^{2}H$ value", "Temperature value", and "Salinity value".

- *Level 4* metadata fields also add important supplementary information related to the seawater $\delta^{18}O$ data, but they are not generalizable enough to use standardized vocabularies. They are also generally not available for all records and thus not required for inclusion in the database. They were subject to the highest QC standard. Examples of Level 4 metadata are: "$\delta^{18}O$ correction notes", "$\delta^{18}O$ error notes", "Sample ID", "Publication citation", "Dataset citation", "Cruise ID", "$\delta^{18}O$ analysis location", "Sample collection, processing, and storage notes", and "Water isotope analysis notes".

- *Level 5* metadata fields may be useful to some users of the database but they are generally not available for all records and thus not required for inclusion in the database. In many cases, these fields contain freeform text with direct quotes from the original publications. During the QC certification process, these fields were checked against the original publication and a quote or summary of the relevant information was provided in the database, but the information provided may not be comprehensive. Examples of Level 5 metadata are: "Location description" and "Location type".

- *Level 6* metadata fields may be useful to some users of the database. This metadata field was completed when the information was easily accessible from the original publications, but some metadata may be missing. There is only one Level 6 metadata field in the database: "Temperature/salinity notes".

**Table 1: Description of all metadata fields in the PAGES CoralHydro2k Seawater $\delta^{18}O$ Database. Bold text indicates required fields in the database (Level 1 and 2).**

| Level # | Metadata field | Metadata field description | Type | Metadata category |
|---|---|---|---|---|
| 1 | **CoralHydro2kID*** | **Unique ID for this database** | **Text** | **Entity** |
| 1 | **Collection year** | **Year of sample collection, YYYY** | **Numeric** | **Entity** |
| 1 | **Collection month** | **Month of sample collection, MM** | **Numeric** | **Entity** |
| 1 | **Latitude** | **Latitude of the sampling site in decimal degrees. South is negative. Decimal degrees N, from -180 to 180** | **Numeric** | **Entity** |
| 1 | **Longitude** | **Longitude of the sampling site in decimal degrees.** | **Numeric** | **Entity** |



| | | West is negative. Decimal degrees E, from -180 to 180 | | |
|---|---|---|---|---|
| 1 | **Depth** | **Sampling depth, in meters (m) below sea level (no minus sign).** | **Numeric** | **Entity** |
| 1 | **Depth units** | **Depth measurement units (meters below sea level)** | **Text** | **Entity** |
| 1 | **δ¹⁸O value** | **Measured δ¹⁸O value** | **Numeric** | **Seawater Data** |
| 1 | **δ¹⁸O units** | **δ¹⁸O units (per mille)** | **Text** | **Seawater Data** |
| 1 | **δ¹⁸O correction** | **Indicates whether a correction has been made to the δ¹⁸O values after the original publication. If the data point has been corrected, "Y" is indicated. If the data has not been corrected, "N" is indicated. If known (e.g., in the data collected from the NASA GISS δ¹⁸O$_{sw}$ database), the value of the applied correction is indicated in the "δ18O correction notes" metadata field. Additional information about each correction is provided in "δ18O correction notes" (Level 4) metadata field.\*\*** | **Logic** | **Seawater Data** |
| 1 | **Evaporation flag** | **Flag indicating the presence of potential evaporation effects on δ¹⁸O value. "Y" is indicated for cases where authors note that sample evaporation could be a concern or cases where δ¹⁸O data have been corrected for evaporation; "N" otherwise. Further information is provided in the "Sample collection, processing, and storage notes" (Level 4) metadata field.** | **Logic** | **Queryable** |
| 1 | **Reference standard** | **Reference standard used in reporting the δ¹⁸O and δ²H values (SMOW, VSMOW)** | **Text** | **Seawater Data** |
| 1 | **Access date** | **Date in which the data was downloaded from data repositories, submitted by researchers, or acquired from journal articles (YYYY/MM/DD)** | **Text** | **Entity** |
| | | | | |
| 2 | **Site name or geographic area** | **Name of the site or the general area from which the water sample was collected** | **Text** | **Entity** |
| | | | | |
| 3 | Collection day | Day of sample collection, DD. In some cases, the collection day was not specified in the original publication, only the collection month or a range of dates. In these cases, the midpoint of the date range | Numeric | Entity |



| | | was selected as the collection day and note is made in the 'Collection date notes' (Level 4) metadata field. | | |
|---|---|---|---|---|
| 3 | Collection time | Time of sample collection, in Coordinated Universal Time (UTC) 24-hour format, HH:MM:SS | Text | Entity |
| 3 | Water isotope analysis date | Date of water isotope analysis (YYYY/MM/DD) | Text | Entity |
| 3 | $\delta^{18}O$ error | Reported error of the $\delta^{18}O$ value. Because many different types of error are reported in the literature, standardization was impossible; therefore, we report the most comprehensive error provided. The type of error along with any supporting information is provided in the "δ18O error notes" (Level 4) metadata field. | Numeric | Seawater Data |
| 3 | $\delta^{18}O$ error units | If $\delta^{18}O$ error exists, the units are specified (per mille) | Text | Seawater Data |
| 3 | $\delta^{18}O$ analysis technique | Type of instrument used to make the isotope measurement (isotope ratio mass spectrometry (IRMS), Cavity Ring Down Spectroscopy (CRDS), and off-axis integrated cavity output spectroscopy (ICOS) | Text | Entity |
| 3 | $\delta^{2}H$ value | Measured $\delta^{2}H$ value | Numeric | Seawater Data |
| 3 | $\delta^{2}H$ units | $\delta^{2}H$ units (per mille) | Text | Seawater Data |
| 3 | $\delta^{2}H$ error | Reported error of the $\delta^{2}H$ value. Because many different types of error are reported in the literature, standardization was impossible; therefore, we report the most comprehensive error provided. | Numeric | Seawater Data |
| 3 | Temperature value | Seawater temperature (degrees Celsius) | Numeric | Seawater Data |
| 3 | Temperature units | Units of temperature value (degrees Celsius). | Text | Seawater Data |
| 3 | Temperature error | Reported error of the temperature value | Numeric | Seawater Data |
| 3 | Salinity value | Seawater salinity | Numeric | Seawater Data |
| 3 | Salinity units | Salinity units (typically "PSU" or "parts per thousand"). Retain original units provided in the publication. | Text | Seawater Data |

| 3 | Salinity error | Reported error of the salinity value | Numeric | Seawater Data |
|---|---|---|---|---|
| | | | | |
| 4 | Collection date notes | A note is made here if the collection day was not specified in the original publication, and only the collection month or a range of dates were specified. In this case the midpoint of the date range was selected as the collection day. | Text | Entity |
| 4 | Location notes | A note is made here if the latitude and longitude coordinates are not exact (e.g., some of the NASA GISS database entries have notes stating "A: Position was read off a graph of locations and therefore is not exact"). In these cases, the notes are copied to this metadata field. | Text | Entity |
| 4 | Depth notes | If only pressure (and not depth) was reported in the original dataset, a note is made here about how the depth conversion was performed. If a range of depths is provided in the original dataset, the midpoint of the depth range is reported in the 'Depth' field and the range is stated here. | Text | Entity |
| 4 | $\delta^{18}$O correction notes | If any $\delta^{18}$O correction was made to the $\delta^{18}$O values subsequent to the original publication, the value of the correction is reported here, along with any accompanying information about how and why the correction was made. | Text | Seawater Data |
| 4 | $\delta^{18}$O error notes | Information about the reported error of the $\delta^{18}$O value, including the type of error along with any supporting information. | Text | Seawater Data |
| 4 | Sample ID | Unique sample ID provided by original authors. | Text | Queryable |
| 4 | $\delta^{18}$O analysis location | University or institute where the isotope measurements were made. | Text | Entity |
| 4 | Publication citation | Citation of the original publication of the data. When the data was obtained from a data repository and no publication citation was found, "NaN" is entered in this field and the relevant citation appears in the "Dataset citation" field. | Text | Entity |
| 4 | Publication DOI or URL | DOI or URL of the original publication. | Text | Entity |
| 4 | Dataset citation | If the data was obtained from a data repository, the dataset citation is provided here. | Text | Entity |

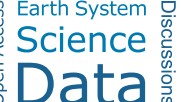

| 4 | Dataset URL | If the data was obtained from a data repository, the dataset URL is provided here. | Text | Entity |
|---|---|---|---|---|
| 4 | Dataset ID | If the data was obtained from a data repository, the Dataset ID from that repository is provided here. | Text | Entity |
| 4 | Data provenance notes | Indicates the source of the $\delta^{18}O$ data (and $\delta^2H$, SST, and SSS data). In some cases, select metadata may have been collected from other data sources, such as the original publication, which is indicated in this field. | Text | Entity |
| 4 | Cruise ID | Cruise ID, if applicable. | Text | Entity |
| 4 | Station ID | Station ID, if applicable. | Text | Entity |
| 4 | Cruise report | Citation of the cruise report, if applicable. | Text | Entity |
| 4 | Cruise report URL | Link to the cruise report, if applicable. | Text | Entity |
| 4 | Sample collection, processing, and storage notes | Notes on sample collection, sample processing, and sample storage. Sample processing includes any treatments prior to analysis (distillation, filtration, etc). Sample storage includes the type of storage container and any preventative measures taken against evaporation (sample seals, refrigeration, etc). | Text | Entity |
| 4 | Water isotope analysis notes | Notes on the isotope analysis methods, including details of any equilibration steps used between the water and $CO_2$ or $H_2$ gas, the specific type of instrumentation used for the isotopic analysis, details of the calibration steps and standards used, number of replicate measurements, corrections for instrumental drift and memory effects, and preventative steps taken to minimize salt contamination (e.g., for cavity ring down spectroscopy). | Text | Entity |
| | | | | |
| 5 | Location type | Type of water body from which the water samples were collected (e.g., open ocean, coastal, bay, lagoon, estuary, enclosed sea, marginal/semi-enclosed sea). | Text | Entity |
| 5 | Location description | Description of the sampling location, including a description of the major water masses and currents influencing the region, as well as details on the surface water balance, groundwater or riverine input, upwelling, distance from the coastline, depth and geometry of bay or lagoon, and/or description of co-located coral reef site if applicable. | Text | Entity |
| | | | | |



| 6 | Temperature/salinity notes | Notes on the analysis of temperature and salinity. | Text | Entity |
|---|---|---|---|---|

*There are four cases in which datasets obtained from the GISS database were not clearly associated with a publication, or the provided reference did not match the dataset. In those cases, the Schmidt et al. (1999) database citation is provided in the

225 "Dataset citation" metadata field and the author letters "SC" are used in the unique CoralHydro2k ID to reference that citation (SC99AO0001, SC99PO0001, SC99IO0001, SC99GI0001). Additional details about the citations and data provenance appear in the "Data provenance notes" metadata field.

**Corrections were applied in several datasets in the NASA GISS $\delta^{18}O_{sw}$ database to standardize the data based on deep water masses to correct for changes in standards, different analysis techniques, and other systematic errors. All corrections are noted

230 in the "δ18O correction" metadata field, so the user can remove the corrections if desired. Corrections were also applied in some datasets in the Reverdin et al. (2022) LOCEAN database to adjust for minor evaporation biases. In that case, the correction value is unknown.

**Table 2: Standardized controlled vocabulary options for metadata fields in the database.**

| Metadata Field | Standardized Entries |
|---|---|
| Depth units | m (below sea level) |
| $\delta^{18}O$ units | per mille |
| $\delta^{18}O$ correction | Y, N |
| Evaporation flag | Y, N |
| Reference standard | VSMOW, SMOW |
| $\delta^{18}O$ error units | per mille |
| $\delta^{18}O$ analysis technique | IRMS, CRDS, ICOS |
| $\delta^2H$ units | per mille |
| Temperature units | °C |
| Salinity units | PSU, PSS-78, PPT, g/L |

Earth System
Science
Data

Open Access | Discussions

## 3 Key characteristics of the seawater δ¹⁸O data

### 3.1 Spatial and temporal coverage of the database

The CoralHydro2k Seawater $\delta^{18}O$ Database contains 18,598 data points from 106 datasets (Fig. 1A,B). 53% of the data is categorized as "hidden" data (i.e., data not currently available in public databases or public repositories; Fig. 1C,D), and the remaining 47% of the data is from public databases or public repositories (Table 3). 10,407 measurements (56%) are from the sea surface (depth ≤ 5 m; Fig. 1A), 3,693 (20%) are from the mixed layer (between 5–50 m), and 4,498 (24%) are below 50 m. The time span of the database covers 1972 to 2021 (Fig. 2) and the depth range covers the surface to 5,797 m below sea level. The earliest data point in the database was collected in September 1972 and the most recent data point was collected on October 8, 2021. 3,480 data points (19%) were collected before the year 2000, and 15,118 data points (81%) were collected on or after the year 2000 (Fig. 2). Because the search for hidden datasets focused on the region between 35ºN and 35ºS, 75% of the measurements in the database are located within the tropical-subtropical region.

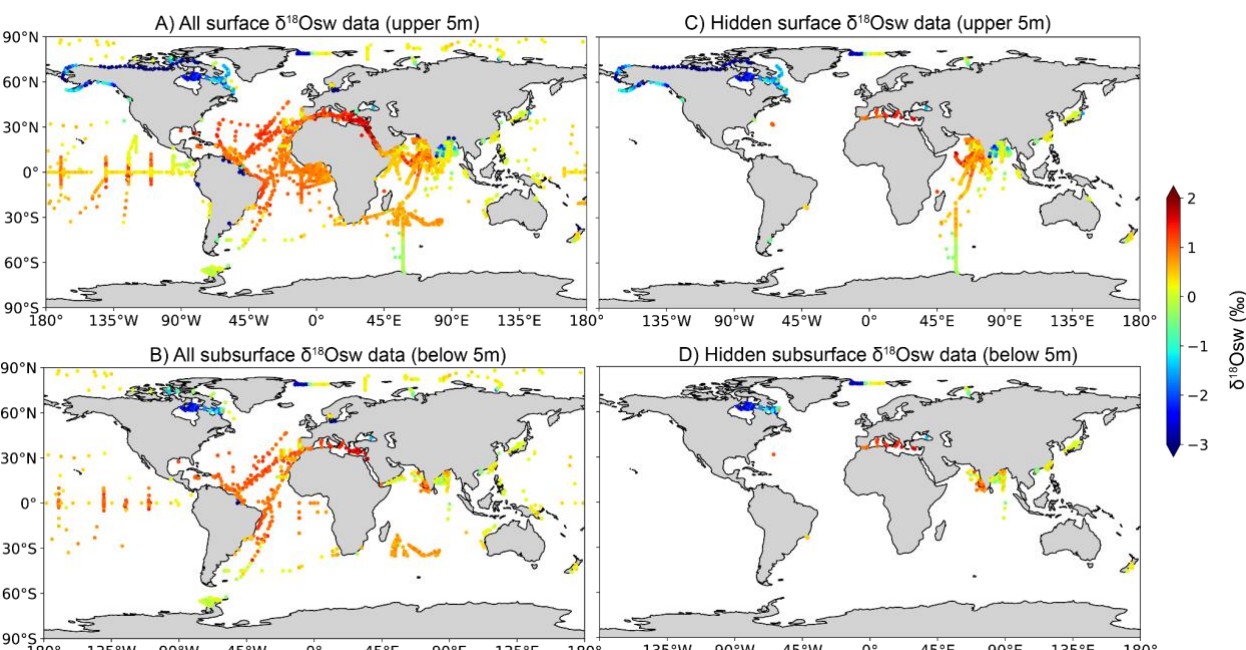

**Figure 1: Locations and seawater δ¹⁸O values of data in the database: (A) all surface ocean data (upper 5 m of the water column), (B) all subsurface ocean data (below 5 m), (C) the hidden surface ocean data only (upper 5 m), (D) the hidden subsurface ocean data only (below 5 m).**

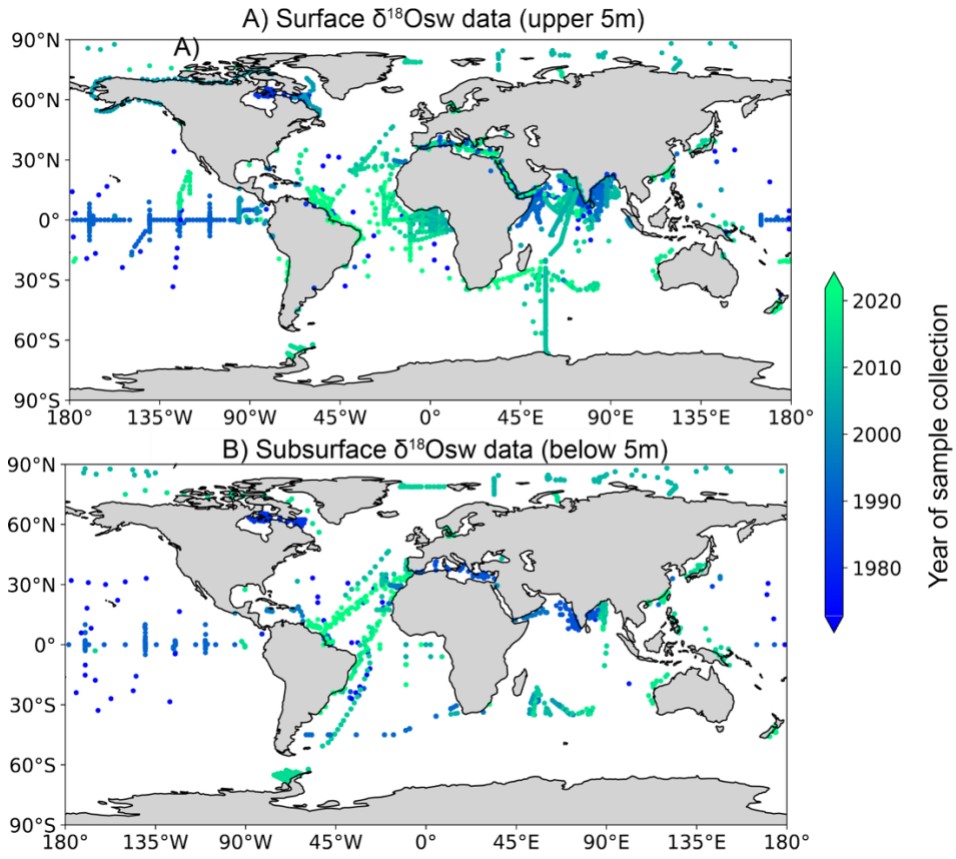

**Figure 2: Sample collection dates of the seawater δ¹⁸O data in the database from (A) the surface ocean (upper 5 m) and (B) the subsurface ocean (below 5 m).**

255 In addition to $\delta^{18}$O measurements, the database also includes paired $\delta^2$H, salinity, and temperature measurements when available. 16,098 data points (87%) have paired salinity values (Fig. 3A), 13,871 data points (75%) have paired temperature values (Fig. 3B), and 9,769 data points (53%) have paired $\delta^2$H measurements (Fig. 3C). 185 measurements have an evaporation flag (Fig. 3D), which allows the user to filter out samples that may be influenced by post-collection evaporation from the database.





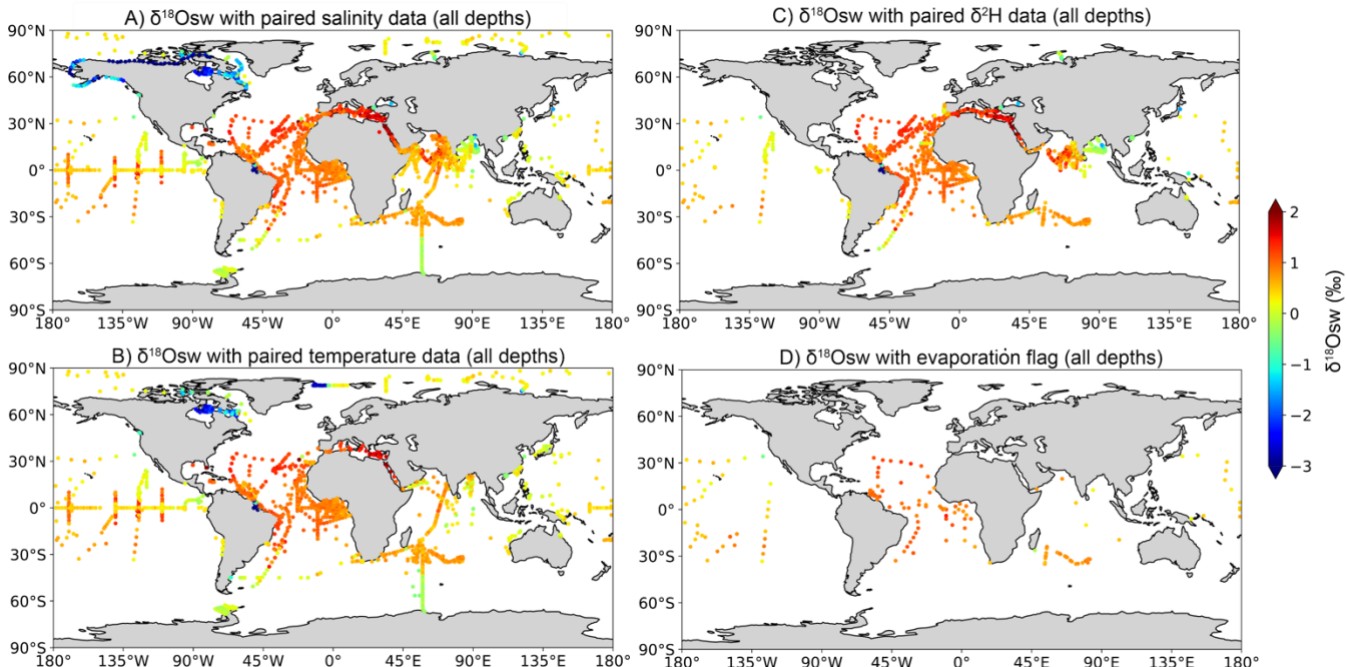

**Figure 3: Locations and seawater δ¹⁸O values of the data in the database with (A) paired salinity measurements, (B) paired temperature measurements, (C) paired δ²H measurements, and (D) an evaporation flag.**

Compared to the seawater $\delta^{18}O$ database presented in LeGrande and Schmidt (2006), data coverage in the surface ocean (upper 5 m) is substantially improved in the tropics and subtropics, particularly in the northern Indian Ocean, the eastern Atlantic Ocean, the northeast coast of South America, the Mediterranean Sea, and the equatorial Pacific Ocean. However, poor data coverage still exists in the western Pacific Ocean and Maritime Continent region, the southeastern Indian Ocean, and the subtropical Pacific Ocean regions in both hemispheres. Below 5 m depth, the data coverage is even more limited (Fig. 1B). At all depths, regions with reasonable spatial coverage of $\delta^{18}O_{sw}$ data contain limited temporal coverage. Typically, only a few years of regular measurements are available from the most highly sampled regions. For example, only 13% of locations contain at least 12 measurements spanning two years within a 2º latitude x 2º longitude grid box (Fig. 4A), and only 9% of the grid boxes contain data that cover at least 50% of the annual cycle (i.e., 6/12 months of the calendar year; Fig. 4B). While the coverage of seawater isotope data has been growing over the last decade, these measurements are still sparse in space and time, thus highlighting the need for globally coordinated sampling campaigns and archiving efforts.



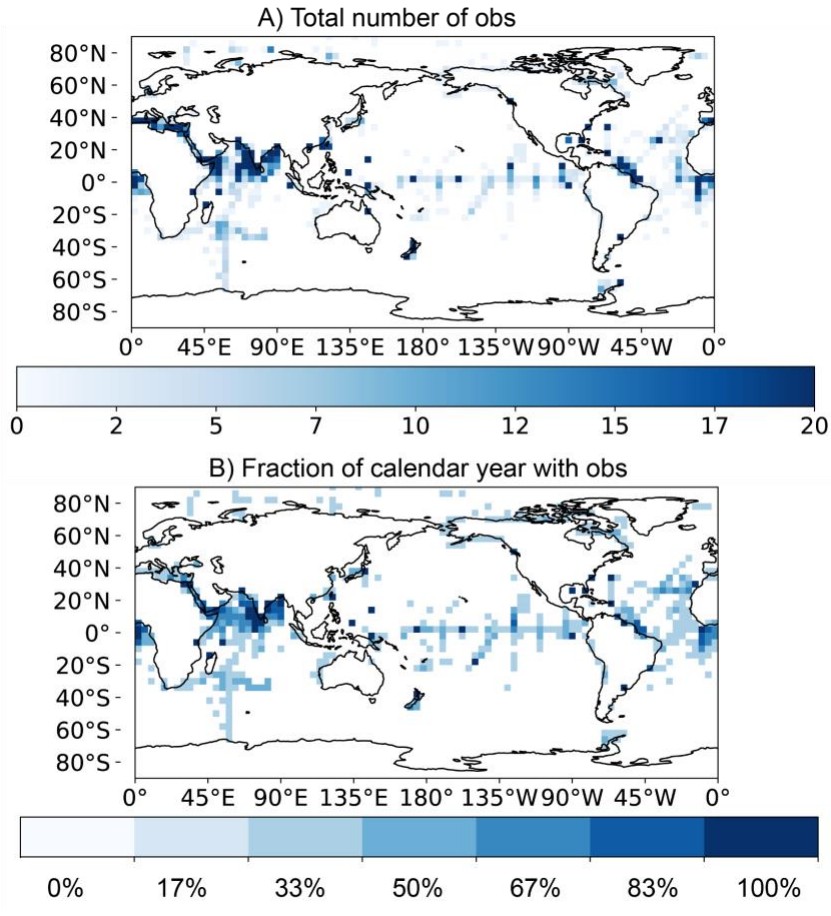

**Figure 4: Temporal distribution of near-surface (upper 5m) δ18Osw measurements in the database, aggregated in 2°x2° (lat x lon) grid boxes. (A) Total number of δ18Osw measurements in each grid box. (B) Fraction of calendar year with δ$^{18}$Osw measurements.**

Outside of the tropics and subtropics, the coverage of $\delta^{18}O$ data in the CoralHydro2k Seawater $\delta^{18}O$ Database is more sparse, since only hidden datasets were collected from all latitudes and all depths across the global ocean, while datasets from public repositories were only incorporated into the database if the measurements were made in the upper 50 m between 35°N and 35°S. Future database development efforts will include incorporating additional hidden and public datasets.

## 3.2 Data-model comparisons of the seawater $\delta^{18}O$ data

To assess how the $\delta^{18}O_{sw}$ data in the database compares with isotope-enabled climate model simulations and other products, we compare the climatological annual cycle in $\delta^{18}O_{sw}$ at different island sites using four data products: two simulations of isotope-enabled General Circulation Models [the National Center for Atmospheric Research Community Earth System Model Last Millennium Ensemble (1000 years; Brady et al., 2019) and the NASA Goddard Institute for Space Studies E2-R last millennium simulation (ensemble member E4rhLMgTck; 255 years; Colose et al., 2016), a regional ocean model of the Pacific



called isoROMS (44 years; Stevenson et al., 2018), and a gridded dataset of global monthly mean $\delta^{18}O_{sw}$ based on data
assimilation with the MITgcm (Breitkreuz et al., 2018). The Breitkreuz dataset is based on a 400-year quasi-equilibrated
simulation of a water isotope-enabled global ocean general circulation model constrained by global monthly $\delta^{18}O_{sw}$ data
collected from 1950 to 2011 and climatological salinity and temperature data collected from 1951 to 1980.

The characteristics of $\delta^{18}O_{sw}$ variability at the four selected sites in the tropical Pacific and Atlantic Oceans vary widely across
the different data products and the $\delta^{18}O_{sw}$ observations from the CoralHydro2k Seawater $\delta^{18}O$ Database, with large differences
in both the amplitude and phase of the annual cycle of $\delta^{18}O_{sw}$ (Fig. 5). These differences could be due to deficiencies in the
models (associated with model resolution, subgrid-scale parameterizations, and treatment of atmospheric exchange or ocean
mixing processes), and/or uncertainties in the observational data given the low temporal resolution of the $\delta^{18}O_{sw}$ measurements.
Clearly, more observational data is needed to determine the source of the discrepancies, pointing to the need for more
coordinated and sustained seawater isotope sampling programs. Seawater isotope sampling at multinational observing systems
that are already in place, such as the Tropical Pacific Observing System (TPOS), Bermuda Atlantic Time-series Study (BATS),
GO-SHIP, and GEOTRACES, could expand and complement existing observational programs. For example, incorporating
new sampling devices such as long-term osmotically pumped fluid samplers (Jannasch et al., 2004; Khare et al., 2021) could
provide a relatively straightforward way to add $\delta^{18}O_{sw}$ measurements to existing programs. The development of sustained
seawater isotope measurements at a network of observational hotspots around the global ocean would provide powerful new
constraints on hydrologic changes in the modern ocean, generating data that could be used to test theoretical predictions, assess
climate model performance and skill, and calibrate paleoclimate proxies for improved paleoclimate reconstruction.

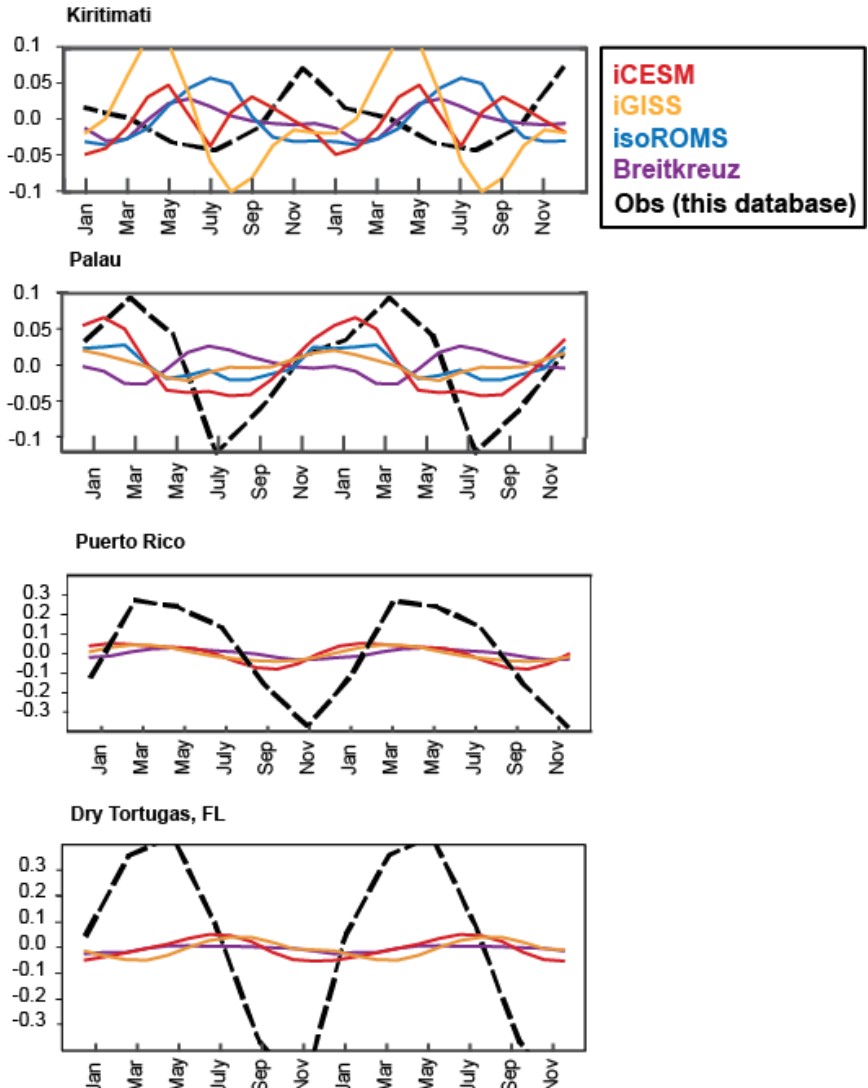

**Figure 5: Monthly climatology of δ¹⁸Osw at four island locations: Kiritimati Atoll in the Central Pacific Ocean, Palau in the western**
**Pacific Ocean, Puerto Rico in the Carribbean Sea, and Dry Tortugas in the Gulf of Mexico. Five data sources are shown: observed δ18Osw (from this database; black dashed line), simulated δ¹⁸Osw from iCESM, iGISS, and isoROMS, and δ¹⁸Osw from the reanalysis product of Breitkreuz et al., 2018 (purple; monthly climatology constrained by observed monthly δ18Osw data collected from 1950 to 2011 and climatological salinity and temperature data collected from 1951 to 1980). The three Earth system models are the National Center for Atmospheric Research Community Earth System Model Last Millennium Ensemble (1,000 years; red)**
**(Brady et al., 2019), the NASA Goddard Institute for Space Studies E2-R last millennium simulation (ensemble member E4rhLMgTck; 255 years; yellow) (Colose et al., 2016), and the isoROMs Pacific Ocean simulation (44 years; blue) (Stevenson et al., 2018).**



## 4 Usage notes

### 4.1 General applications

While the primary motivation of the CoralHydro2k Seawater $\delta^{18}$O Database was for coral paleoclimate research, this database was designed to be useful to researchers from a wide range of disciplines, including paleoceanography and paleoclimatology, oceanography, marine biology, Earth science, and climatology. For example, of relevance to oceanography, the seawater $\delta^{18}$O data can be coupled with the paired salinity data in the database to explore the relationship between these two parameters and investigate hydrological changes of the surface ocean (e.g., Conroy et al., 2017; Durack et al., 2012; LeGrande and Schmidt,

2006; Wagner and Slowey, 2011). Because this relationship varies with latitude and may vary with time (Conroy et al., 2017; LeGrande and Schmidt, 2011; Thompson et al., 2022), the seawater $\delta^{18}$O database could be used to more comprehensively assess how this relationship varies in space and time.

As for applications in paleoclimatology, the pairing of seawater oxygen isotope data with salinity data can provide transfer

equations for reconstructing past salinity variations (e.g., Gagan et al., 1998; Kilbourne et al., 2004; McCulloch et al., 1994; Ren et al., 2003). For example: coral Sr/Ca, a temperature proxy, paired with coral $\delta^{18}$O, a proxy for both temperature and $\delta^{18}$O$_{sw}$, can be used to remove the temperature component from the coral $\delta^{18}$O signal. The derived $\delta^{18}$O$_{sw}$ can then be converted to salinity using the local $\delta^{18}$O$_{sw}$ to salinity transfer equation (e.g., Kilbourne et al., 2004). The same method can be applied to foraminifera Mg/Ca and $\delta^{18}$O records to reconstruct salinity variations and can also potentially be applied to bivalves, coralline

algae, ostracods, and otoliths (e.g., Light et al., 2018; Schmidt and Lynch-Stieglitz, 2011; Stott et al., 2004; Trofimova et al., 2020; Warner et al., 2022). Many studies have used this paired approach to reconstruct $\delta^{18}$O$_{sw}$ variations for a wide range of time scales (e.g., Brocas et al., 2019; Felis et al., 2009; Giry et al., 2013; Gorman et al., 2012; Hereid et al., 2013; Knebel et al., 2024; Wu et al., 2013); however, few studies have been able to validate their reconstructions with observed $\delta^{18}$O$_{sw}$ records that span more than one year (Conroy et al., 2017; O'Connor et al., 2021). Instead, most studies use reanalysis products such

as Simple Ocean Data Assimilation (SODA) (Carton et al., 2000, 2018), or satellite-derived sea surface salinity (SSS) products (e.g., NASA Aquarius, NASA SMOS) (Boutin et al., 2021) for validating the reconstructions (e.g., Cahyarini et al., 2008; Harbott et al., 2023; Hetzinger et al., 2006).

Additionally, the CoralHydro2k Seawater $\delta^{18}$O Database can be used in proxy-system model development, paleo-data

assimilation, and comparison studies between proxy reconstructions and climate model output (Dee et al., 2023; Evans et al., 2013; Reed et al., 2022; Sanchez et al., 2021; Smerdon, 2012; Stevenson et al., 2018; Thompson et al., 2011). Proxy-derived $\delta^{18}$O$_{sw}$ data can be directly compared with simulations from isotope-enabled models as part of the validation process and to understand oxygen isotope fractionation processes within the hydrological cycle (Dee et al., 2015; Stevenson et al., 2015, 2023). Furthermore, the proxy-derived salinity reconstructions can be compared with reanalysis and other salinity data

products, such as SODA, as a separate validation step (e.g., Cahyarini et al., 2008). Finally, the $\delta^{18}$O$_{sw}$ database offers the





opportunity for improved proxy system models with rigorous uncertainty quantification of proxy-derived estimates of salinity. With such estimates, long reconstructions of salinity would provide valuable insights into the low frequency variability of the hydrological cycle over the data sparse tropical oceans. This $\delta^{18}O_{sw}$ database is the most comprehensive to date and will be updated as new datasets are published to support ongoing research (see Section 6).

**5 Code/Data availability**

**5.1 Accessing the database**

The CoralHydro2k Seawater $\delta^{18}O$ Database follows the FAIR data principles (Wilkinson et al., 2016) that strive to make scholarly data findable, accessible, interoperable, and reusable. The CoralHydro2k Seawater $\delta^{18}O$ Database uses the Comma Separated Values (*.CSV) file format, a machine-readable format for archiving and describing seawater isotope data. Access 360 to the database has been granted for reviewers and editors during the review phase. The data are also available upon request for members of the public that wish to participate in the review process by emailing the corresponding author. Once the review period is complete, the database will be archived on the NOAA NCEI World Data Service for Paleoclimatology (study page: https://www.ncei.noaa.gov/access/paleo-search/study/34575) and issued a permanent DOI. A mirror copy of the database will also be hosted at Waterisotopes.org.

**5.2 Code availability**

Example scripts to help users filter and search the database are available on the CoralHydro2k Seawater Database GitHub page (https://github.com/CoralHydro2k/ch2kSeawater_Database).

**5.3 Underlying data sources**

The CoralHydro2k Seawater $\delta^{18}O$ Database includes records (0–50 mbsl, 35ºN to 35ºS) from ten international databases, 370 including the NASA GISS Global Seawater Oxygen-18 database (Table 3). Literature searches were also conducted to find hidden seawater $\delta^{18}O$ data (from all depths and latitudes) published only in tables and supplemental data files of published papers, theses, and dissertations. Data was also sourced from author contributions sent directly to this project or the EarthChem community (earthchem.org/communities/seawater-oxygen-isotopes). Researchers should adhere to the data use policies for the underlying data sources (see Table 3 and Appendices A1–A4).


**Table 3. Databases included**

| Database | URL (https://) | Notes | References |
| --- | --- | --- | --- |

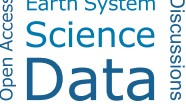

| British Oceanographic Data Centre | https://www.bodc.ac.uk/ | See A1 for data usage policies | Acknowledge the source of the information by including any attribution statement, see A4. |
|---|---|---|---|
| CISE LOCEAN Seawater Isotope (SEANOE) | www.seanoe.org/data/00600/71186/) | Public domain, see URL for data usage details. | (waterisotopes-CISE-LOCEAN 2024), (Reverdin et al. 2022) |
| GEOTRACES | geotraces.webodv.awi.de/ | See A2 for data usage policies | (GEOTRACES Intermediate Data Product Group, 2023) |
| Global Ocean Data Analysis Project (GLODAPv2.2022) | www.ncei.noaa.gov/access/metadata/landing-page/bin/iso?id=gov.noaa.nodc:0257247 | See A3 for data usage policies | (Key et al., 2023; Olsen et al., 2016) |
| NASA GISS Global Seawater Oxygen-18 (GISS) | data.giss.nasa.gov/o18data/ | Public domain | (Bigg and Rohling, 2000; Schmidt, 1999; Schmidt et al., 1999) |
| NOAA National Centers for Environmental Information (NCEI) | www.ncei.noaa.gov/ | Public domain | Cite original publication, online resource, dataset and publication DOIs (where available), and date accessed |
| NOAA NCEI Paleoclimatology | www.ncei.noaa.gov/products/paleoclimatology | Public domain | Cite original publication, online resource, dataset and publication DOIs (where available), and date accessed |
| NOAA NCEI World Ocean | www.ncei.noaa.gov/products/world-ocean-database | Public domain | (Boyer et al., 2018) |
| PANGAEA | www.pangaea.de/ | Terms of use https://www.pangaea.de/about/terms.php | (Felden et al. 2023) |
| Waterisotopes.org (WIDB) | https://wateriso.utah.edu/waterisotopes/ | See A4 for data usage policies | http://waterisotopes.org |



## 6 Database submission of new datasets and versioning scheme

The CoralHydro2k Seawater $\delta^{18}$O project will accept data submissions for updates to the database. All seawater $\delta^{18}$O observations are welcomed regardless of location or water depth. To facilitate this process, a Seawater Oxygen Isotopes
Community was developed within the EarthChem Library, an open-access repository for geochemical datasets (earthchem.org/communities/seawater-oxygen-isotopes), where researchers can submit their seawater isotope data and obtain a dataset DOI. The Seawater Oxygen Isotopes Community contains a template that can be downloaded to help researchers submit their data (scroll to the bottom of the webpage above and click on "Download Template"). This template is aligned with the CoralHydro2k Seawater $\delta^{18}$O Database to facilitate future updates to the database. The template has a README tab
in the Microsoft Excel file with details on the template and an example. We hope that the creation of this site helps researchers publish their seawater isotope datasets, thus minimizing the number of "hidden" datasets.

The initial release of the CoralHydro2k Seawater $\delta^{18}$O Database will be Version 1.0.0 for this publication. With new submissions, the database will grow as new datasets are added. Database users who find errors in the database can use the
"Report an issue" option in the GitHub site. Datasets submitted to the Seawater Oxygen Isotopes Community within the EarthChem Library (earthchem.org/communities/seawater-oxygen-isotopes) can be updated through that site.

As the CoralHydro2k Seawater $\delta^{18}$O Database is updated, it will be versioned following the scheme used by other PAGES data collection projects (Ahmed et al., 2013; Emile-Geay et al., 2017; Kaufman et al., 2020; McKay and Kaufman, 2014;
Walter et al., 2023). The version number has three counters in the following form: $C_1.C_2.C_3$, where $C_1$, $C_2$, and $C_3$ are incrementing integers. When $C_1$ increases, $C_2$ and $C_3$ reset to zero. When $C_2$ increases, $C_3$ resets to zero. $C_1$ represents the number of publications describing the database. $C_2$ increments each time the set of records in the database changes (addition or removal of a dataset). $C_3$ increments when the data or metadata within the dataset changes, but the set of records remains the same. Upon updates, extensions, or corrections to the database, rather than issuing errata to this publication, changes will
be included in subsequent versions of the database and updated and described through the online data repository.

## 7 Citation

This CoralHydro2k Seawater $\delta^{18}$O Database descriptor publication should be cited when the database is used in whole or in part, including its metadata fields, for subsequent studies. We encourage end users of this database to also cite the original publications and/or data sources of the underlying primary data (Table 3). To facilitate this process, citation information for
every data point is included in the metadata, including a full citation and DOI of the original publication, as well as a dataset citation and DOI for the original public archive of the data. Researchers should also adhere to the data use policies for the underlying data sources (see Appendices A1–A4).



## 8 Conclusions and anticipated applications of the Seawater $\delta^{18}$O Database

Observational seawater $\delta^{18}$O and $\delta^2$H data can place powerful constraints on the global water cycle, providing valuable
information on the exchange of water between the ocean, atmosphere, and cryosphere, as well as on ocean-mixing processes.
As such, these data provide an additional degree of freedom for understanding the complex hydrologic system, beyond what
standard oceanographic variables like temperature and salinity can offer. They also provide a "common currency" that links
paleoclimate reconstructions, modern climate observations, and isotope-enabled model simulations, allowing hydrologic
processes to be evaluated on a wide range of time and spatial scales. Given the broad value of this data, and the growing
number of seawater $\delta^{18}$O and $\delta^2$H datasets that have been generated since 2011, the CoralHydro2k Seawater $\delta^{18}$O Database
was developed to improve the accessibility of seawater isotope data for the Earth Science research community. This new,
machine-readable, and metadata-rich database contains over 18,600 observational seawater $\delta^{18}$O data points, paired with
seawater $\delta^2$H and salinity data and extensive metadata that makes the database suitable for a myriad of research applications.
The metadata template also provides a set of best practices for reporting seawater isotope data in future studies.


The CoralHydro2k Seawater $\delta^{18}$O Database and its extensive metadata can provide insight into the multiple processes that
impact seawater $\delta^{18}$O and $\delta^2$H. Furthermore, the database can be used to better constrain the relationship between $\delta^{18}O_{sw}$ and
salinity in the global ocean, and assess how this relationship varies in space and time. The database also provides updated
seawater $\delta^{18}$O and $\delta^2$H data critical for the calibration and validation of paleoclimate reconstructions using $\delta^{18}$O and $\delta^2$H to
reconstruct past ocean temperature and salinity variations. For example, recent paleoclimate data assimilation efforts would
greatly benefit from a spatial network of observational $\delta^{18}O_{sw}$ data for training the proxy system models that underlie those
efforts. This database could also be used to construct a new gridded dataset of $\delta^{18}O_{sw}$ to update that of (LeGrande and Schmidt
2006), which has been widely used for providing climate model boundary conditions and to assess model performance and
skill in resolving key features of the hydrologic cycle. In this way, the PAGES CoralHydro2k Seawater $\delta^{18}$O Database can be
used in a wide variety of applications to bolster our understanding of the modern climate system, while also providing new
insights into past and future climate variability and change.

## 9 Author contribution

AA, KD, AM, TF, SL, SS, and ED designed the database, AA, AM, RP, SL, KD, JH, CM, SS, and AV entered data and/or
metadata into the database, AA, AM, SL, RP, JH, CM, and ED performed quality control on the database, AA and KD prepared
the manuscript, with contributions from all co-authors, RP developed the example Python code, and JH developed the Github
site for the database.

## 10 Competing interests

The authors declare that they have no conflict of interest.



## 11 Acknowledgements

We gratefully acknowledge the many researchers, funding agencies, and international project teams responsible for the collection, quality control, and publication of the seawater isotope data, including GLODAP and GEOTRACES. We thank the PAGES CoralHydro2k team for their encouragement and effort in building this database, especially the helpful comments and suggestions from Amy Wagner and Hali Kilbourne. We also thank Gilles Reverdin for useful discussions about the database. Many thanks to Erika Ornouski for her work in finding hidden $\delta^{18}O_{sw}$ data files in the early stages of the project. We are grateful to Kerstin Lehnert and the EarthChem team at Lamont Doherty Earth Observatory and Carrie Morrill and Edward Gille and Bruce Bauer at NOAA/WDS Paleoclimatology for providing opportunities to host the new database. We also recognize the efforts of Gavin Schmidt, Eelco Rohling, Grant Bigg, and Allegra LeGrande in building and maintaining the first seawater $\delta^{18}O$ database, as well as Gabriel Bowen at waterisotopes.org and Gilles Reverdin at LOCEAN-IPSL (https://www.locean.ipsl.fr/?lang=fr) for their data collection and compiling efforts in continuing to make such data publicly available.

The GEOTRACES 2021 Intermediate Data Product version 2 (IDP2021v2) represents an international collaboration and is endorsed by the Scientific Committee on Oceanic Research (SCOR). The many researchers and funding agencies responsible for the collection of data and quality control are thanked for their contributions to the IDP2021v2. This manuscript also contains data supplied by the Natural Environment Research Council.

PAGES provided funding support for this project through the PAGES Data Stewardship Scholarship (DSS_104 and DSS_114 awarded to A.R.A). Additional support for this research came from the National Science Foundation awards OCE-1903640 to A.R.A, OCE-2303245 to S.A.M., OCE-2303565 to S.C.S., and NSF-2102931 to K.L.D. and the Department of the Interior South Central Climate Adaptation Science Center Cooperative Agreement G19AC00086, Louisiana Board of Regents LEQSF(2021-22)-ENH-DE-05 to K.L.D. J.A.H and T.F. acknowledge Deutsche Forschungsgemeinschaft (DFG, German Research Foundation) – Project number 469906366 (T.F.) – SPP 2299/Project number 441832482.

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
