# Peer review of "The PAGES CoralHydro2k Seawater $\delta^{18}O$ Database: A FAIR-aligned compilation of seawater $\delta^{18}O$ data to uncover 'hidden' insights from the global ocean"

_Earth System Science Data, 2025_

## Author Comment (AC1)

**RC2**: , Alessio Rovere, 30 Oct 2025  reply

Dear Editor,

I have now completed my assessment of the manuscript under consideration.

The PAGES CoralHydro2k Seawater $\delta^{18}$O Database represents a new and comprehensive compilation of seawater stable isotope ($\delta^{18}$O and $\delta^2$H) and salinity data, developed in accordance with the FAIR principles (Findable, Accessible, Interoperable, Reusable). The initiative successfully addresses the lack of a centralized archive and makes previously "hidden" or scattered data accessible—information that is essential to the Earth Science research community. Its main goal is to support the calibration of coral-derived paleoclimate proxies and to enhance understanding of tropical hydrological processes and the performance of isotope-enabled climate models. The database, which includes more than 18,600 measurements collected between 1972 and 2021, constitutes the most extensive synthesis of marine isotope observations currently available. However, the authors rightly note the need for further globally coordinated sampling efforts, as the spatial and temporal coverage of the data remains uneven and incomplete.

Overall, the topic is valuable and worthy of publication. The dataset is impressive, and the manuscript reads well. Nevertheless, I believe a few aspects could be strengthened.

First, the description of the data fields seems too focused on what has been standardized from previous works, rather than providing clear guidelines for how new data should be reported. The authors should make an additional effort to describe how new data submissions should be formatted—perhaps specifying character limits or input rules for certain fields (some entries in the current database appear excessively long).

The metadata fields are meant to do just this- provide a specific set of guidelines for new data reporting. We have added a brief discussion on this point to Section 2.3: Metadata description and quality control (now Section 2.4: Metadata, quality control, and best practices for future data reporting). Only 10 metadata fields have standardized vocabulary (provided in Table 2). The remainder are intended to be free form to provide flexibility and ease for the data submitters, thus lowering the energy barrier for submission.

Second, the database's structure—as a single CSV file rather than a true relational database—is a limitation. This format constrains automated validation (e.g., checking date formats, field lengths, or required entries) and makes the file cumbersome to handle. I am not suggesting that the authors completely redesign the structure at this

stage, but these issues should at least be discussed, particularly regarding data quality control and verification.

We believe a simple CSV is the optimal format for this database, as the intent is to make the database interoperable with any software program and in a format that is familiar to the widest possible user base. However, we agree with the reviewer that automated validation would be a beneficial feature and we will consider future options for integrating a validation tool into EarthChem data submission portal.

Concerning usability, I reviewed the GitHub repository containing the example code. Currently, it provides only basic spatial plotting examples with minimal commenting and no use of Markdown cells to explain the workflow—one of the key advantages of Jupyter notebooks. Enhancing this documentation with better-annotated examples and more diverse use cases would greatly benefit potential users.

Thank you for this feedback. We improved the Jupyter notebook by adding explanatory Markdown cells, clearer code comments, and more annotated spatial plotting examples to better demonstrate how users can explore, filter, and visualize the database. In addition, we added a MATLAB script on Github that imports the $\delta^{18}O$ seawater dataset, and provides users with a clear interface to query, filter, and plot the data. We have also added a sentence to Section 5.2: Code availability in the manuscript to encourage users of the database to share their scripts on Github.

To further improve usability and encourage community contributions, I suggest providing a CSV template for new data submissions, along with a validation script to ensure that all fields are completed correctly and all mandatory information is present. This would enhance both the long-term sustainability of the resource and its adoption by the broader community.

An Excel template has been provided for new data submissions. These templates have been uploaded to the database Github page. While a validation script is not available at this time, we agree that it would be a useful tool. We will look into creating such a script to upload to the database Github page and the EarthChem data submission portal.

I hope these suggestions will help the authors strengthen their work.

Thank you for the very helpful suggestions!

---

## Author Comment (AC2)

**RC1**: ['Comment on essd-2025-467'](), Anonymous Referee #1, 29 Sep 2025  reply

This article presents a new machine readable metadata-rich database of observational seawater δ18O data, paired with seawater δ2H and salinity data, that is compliant with the FAIR standards. This is an important step towards improving the accessibility of seawater δ18O data for the Earth Science research community.

The article thus deserves publication. However, in its present form, the article is very short and contains many repetitions, so a number of points detailed below should be improved before it can be accepted for publication.

**Main comments**

- The abstract mentions that the article provides "a set of best practices to the community for reporting seawater isotope data in the future". However, these best practices are not described in the article. There is only one line (l. 419) mentioning that "the metadata template provides a set of best practices for reporting seawater isotope data in future studies". This is not sufficient. The best practices the authors have in mind should be clearly communicated to the community in a dedicated section of the article.

The metadata fields are intended to provide this set of best practices for future data reporting. To clarify this point, we have added the following paragraph to Section 2.3: Metadata description and quality control (now Section 2.4: Metadata, quality control, and best practices for future data reporting):

*In alignment with FAIR data principles, the Seawater $\delta^{18}O$ Database contains extensive metadata. Eight metadata fields are required, with an additional 44 optional metadata fields that provide important supporting information on the sampling site, sample collection and storage, the isotope analysis method, instrumentation, and error information. Where available, paired seawater $\delta^2H$, salinity, and temperature data are also reported.* ***The full set of required and optional metadata fields in the database are intended to establish a set of best practices for future reporting of seawater isotope data. While we consider many of the optional metadata fields to be essential for proper quality control, inter-comparison, and interpretability across datasets, this information was often not reported in the original datasets and publications. We strongly encourage the inclusion of all metadata fields in future***

*data submissions. To assist researchers with this process, a blank data submission template (with examples) is provided with the database.*

- The article should be reorganized to avoid repetitions. For instance there are many repetitions between the introduction, sections 2.1 and 5.3. Also, the statement that "the seawater δ18O data is paired with seawater δ2H, salinity, and temperature data, where available" is repeated in section 2.2. Please suppress as many repetitions as possible to improve the readability of the article.

Thank you for this feedback. We were not able to find any repetition between section 2.2 "Data aggregation and formatting" and section 5.3 "Underlying data sources". However, we made minor edits to the Introduction and Methods sections to remove all instances of repetition.

- It is unclear why the authors searched for datasets spanning all depths and latitudes for hidden data (l. 183) whereas they focused on the upper 50 m between 35°S and 35°N for published datasets. This seems inconsistent. Please clarify the rationale behind this approach. Also, the sentence l. 244 "Because the search for hidden datasets focused on the region between 35°N and 35°S […]" is in contradiction with the definition of the hidden data domain given l. 183, please clarify.

We have clarified the rationale for this approach in the revised text:

*For hidden data, we searched for and included datasets spanning all depths and all latitudes across the global ocean. For publicly available data__, given the substantial time commitment involved in finding and adding the extensive metadata__, we typically only included data from the upper 50 m between 35ºN to 35ºS (to aid in CoralHydro2k's seawater $\delta^{18}O$ reconstruction studies using $\delta^{18}O$ and Sr/Ca in tropical-subtropical corals). __In subsequent versions of the database, we will target the inclusion of all publicly available datasets.__*

Thank for pointing out the error **in** sentence l. 244. We have revised this sentence to:

*Because the **addition** of **public** datasets focused on the region between 35ºN and 35ºS…*

- In the definition of level 6 metadata (l. 217-219), one does not see the difference between level 6 and level 5 metadata. I am guessing that level 5 metadata refer to water d18O, whereas level 6 metadata refer to secondary variables, like temperature. If so, this should be made clear in the definition of the different metadata levels in section 2.3.

The differences between Level 5 and Level 6 metadata were minor, and thus these metadata fields were merged into one group (Level 5).

- 294: another possible reason for the differences between model outputs and observations, is the local influence of E, P and runoff near the selected sampling sites compared to the open ocean far from islands (i.e. the 'island' or 'continental' effects) that is not well accounted for by the models due to their limited spatial resolution.

Thank you for this comment. We have added this as a possible source of the data/model discrepancy in this sentence (Section 3.2).

- 328-338: all these examples of applications assume that the salinity vs d18Osw relationship is stable in time, but it has been shown that this assumption is not verified in many instances: e.g. this relationship varies between monsoon and non-monsoon seasons (McConnell et al., 2009; Gosh et al., 2013), or in regions affected by sea ice formation and melting (Strain and Tan, 1993), and more generally in case of ocean circulation changes (Rohling and Bigg, 1998). This limitation should be explicitly stated.

Thank you for this comment. We have revised Section 4 (Lines 539-550) to make this point and provide motivation for users of the database to explore these relationships between salinity vs d18Osw in the future.

- 411: The term "degree of freedom" is misused: In statistics, the degree of freedom refers to the number of random variables that cannot be determined or fixed by an equation. The way it is used here seems to imply that a higher degree

of freedom would help define/understand the studied system. Please correct the sentence.

This point is well taken. We have revised this phrase to "these data provide an additional set of constraints…"

- The last part of the sentence "the database can be used to better constrain the relationship between δ18Osw and salinity in the global ocean, and assess how this relationship varies in space and time"(l. 423) is an overstatement as far as paleoceanographic time scales are concerned. It is only true for seasonal variations. Please make this clear.

We have revised this sentence to:

*Furthermore, the database can be used to better constrain the relationship between $\delta^{18}O_{sw}$ and salinity in the global ocean, and* **(in conjunction with future improvements in data coverage) provide insight into** *how this relationship varies in space and time,* **on seasonal to decadal timescales, in a warming climate.**

**More minor comments**
- 43-45: maybe mention that the stable isotopes of water composition is nearly conservative in sea water, when no phase exchange is involved (except for very small contributions related to chemical reactions, mostly on inorganic carbon, silicate, and nitrate cycles).

Added.

- 53-55: the link to marine biominerals and lipids is a bit indirect. It is important to mention, but it would be more appropriate to include it in the next paragraph (similarities with what is needed for paleoclimatic reconstructions).

These two sentences were moved to the subsequent paragraph as suggested.

- 82: in 'addition to in situ atmospheric…', mention 'in situ oceanic…'

This paragraph is solely focused on atmospheric measurements, so we prefer to keep the wording as-is.

- 90-91: for precipitation, evaporation and salinity, there exists a very well structured international infrastructure related to GOOS (in addition to Argo, the asset of TAO and other tropical moorings, drifters, ship-of-opportunity measurements), please modify the sentence accordingly.

This sentence was modified to:

*… via satellite remote sensing, the ARGO (Wong et al., 2020)* **and GOOS (Dexter and Summerhayes, 2010) programs, the TAO/TRITON array, and other moorings, drifters, and ship-of-opportunity measurements)…**

- 91: replace "select" by "selected".

Replaced.

- 231-232: the correction for minor evaporation adjustment applied to some data points of the LOCEAN database is defined in the paper accompanying that database (Reverdin et al., 2022): "When breathing was not too large (resulting in an increase of less than +0.11‰ in d18O), we used the deviation from the expected d-excess relationship to S to estimate an adjusted d18O and dD (Benetti et al., 2017)." Note that this correction method is described in Appendix B of Benetti et al. (2017).

The exact correction values, to the best of our knowledge, were not reported in the Reverdin database. However, the method upon which the corrections were based on was provided. Thus, as suggested by the reviewer, we have added this information to the footnote, along with the reference to Benetti et al. (2017).

- 270-271: Fig. 4A is invoked to support the statement "only 13% of locations contain at least 12 measurements spanning two years within a 2° latitude x 2° longitude grid box". However, Fig. 4A only shows the number of observations per 2°x2° grid cell, independently of their timing. Please clarify.

The figure reference was moved to the previous (and more general) sentence:

*At all depths, regions with reasonable spatial coverage of $\delta^{18}O_{sw}$ data contain limited temporal coverage (Fig. 4A).*

- 3C: the distribution of paired data in the southwestern Indian Ocean seems incomplete: most of the LOCEAN dataset in that region consists in data that include both d18O and d2H (as well as T and S) and cover the period 2008-2024.

Given our policy for data inclusion from public databases (i.e. the $\delta^{18}O$ data must be from the upper 50 m between 35°N and 35°S), only 1,182 out of 2,596 data points (46%) in the LOCEAN database from the Southern Indian Ocean region were included in the CoralHydro2k database. Of the included data, 72% contained paired $\delta^2H$ measurements, which compares well with the full LOCEAN dataset, in which 70% of the data from the Southern Indian Ocean region contained paired $\delta^2H$ measurements.

- Subtitle 4.1 should be removed because there is no section 4.2. One option could be to replace "Usage notes" by "Usage notes: General applications".

Subtitle 4.1 was removed and Section 4 was retitled: "Applications of the database".

- 373-374 and 406-407: what do "researchers" refer to? Are these researchers who wish to add data to the CoralHydro2k database? or researchers who wish to download data from the CoralHydro2k database?

Replaced "researchers" with "users of this database".

- 383: "scroll to the bottom of the webpage above and click on "Download Template"". It does not seem necessary to go into such a level of details in the article.

This information was removed as suggested.

- 450: it seems awkward to provide the website of one laboratory and not of the others. Specifying the city and country or just the country could be enough for all the cited laboratories.

For completeness, the websites of all the other databases have been added, along with the affiliation of Gabe Bowen.

- 453: what is the link between the GEOTRACES 2021 Intermediate Data Product version 2 (IDP2021v2) and the CoralHydro2k Seawater δ18O Database? Please explain.

This database includes data from the GEOTRACES 2021 Intermediate Data Product version 2 (IDP2021v2). According to their fair use policy, it is recommended that users of the IDP2021v2 include the following statement in the acknowledgements:

*"The GEOTRACES 2021 Intermediate Data Product version 2 (IDP2021v2) represents an international collaboration and is endorsed by the Scientific Committee on Oceanic Research (SCOR). The many researchers and funding agencies responsible for the collection of data and quality control are thanked for their contributions to the IDP2021v2."*

This information is also provided in the data use policies provided in Appendix A.

**References**

Benetti, M., Reverdin, G., Lique, C., Yashayaev, I., Holliday, N.P., Tynan, E., Torres-Valdes, S., Lherminier, P., Tréguer, P., Sarthou, G., 2017. Composition of freshwater in the spring of 2014 on the southern Labrador shelf and slope. J. Geophys. Res. Oceans 122, 1102–1121. https://doi.org/10.1002/2016JC012244

Ghosh, Prosenjit, Ramananda Chakrabarti, and S. K. Bhattacharya. "Short-and long-term temporal variations in salinity and the oxygen, carbon and hydrogen isotopic compositions of the Hooghly Estuary water, India." Chemical Geology 335 (2013): 118-127.

McConnell, Martha C., et al. "Seasonal variability in the salinity and oxygen isotopic composition of seawater from the Cariaco Basin, Venezuela: Implications for paleosalinity reconstructions." Geochemistry, Geophysics, Geosystems 10.6 (2009).

Rohling, Eelco J., and Grant R. Bigg. "Paleosalinity and δ18O: a critical assessment." Journal of Geophysical Research: Oceans 103.C1 (1998): 1307-1318.

Strain, Peter M., and Francis C. Tan. "Seasonal evolution of oxygen isotope-salinity relationships in high-latitude surface waters." Journal of Geophysical Research: Oceans 98.C8 (1993): 14589-14598.